# Psychometric Evaluation of the Arabic Version of the Person-Centered Climate Questionnaire: Patient Version (PCQ-P)

**DOI:** 10.3390/healthcare11020190

**Published:** 2023-01-08

**Authors:** Mohammed Aljuaid, Bashayer Al-Sahli, David Edvardsson, Khaled Al-Surimi

**Affiliations:** 1Department of Health Administration, College of Business Administration, King Saud University, Riyadh P.O. Box 11450, Saudi Arabia; 2King Faisal Specialist Hospital and Research Center, Riyadh P.O. Box 11564, Saudi Arabia; 3Sahlgrenska Academy, Institute of Health and Care Sciences, University of Gothenburg, P.O. Box 457, SE-405 30 Gothenburg, Sweden; 4School of Nursing and Midwifery, La Trobe University, Bundoora, Melbourne, VIC 3086, Australia; 5School of Health and Environmental Studies, Hamdan Bin Mohammed Smart University, Dubai P.O. Box 71400, United Arab Emirates

**Keywords:** Arabic version, person-centered care, psychometrics analysis, arab countries, Saudi Arabia

## Abstract

Background: Adopting a person-centered care approach has transformed different healthcare settings worldwide. However, this concept has gained little attention in many Middle Eastern countries, including Saudi Arabia and other Arab countries. This study aimed to evaluate the psychometric characteristics of the Arabic person-centered climate questionnaire—patient version, measuring to what extent the climate of health care settings is perceived as being person-centered. Method: This is a psychometric analysis study. The original validated version of the English Person-Centered Climate Questionnaire—Patient version (PCQ-P) was translated into Arabic and tested among a sample of hospital patients (*n* = 300) in Saudi Arabia using translation and back translation procedures. For psychometric evaluation, statistical analyses of validity and reliability were used, including exploratory factor analysis as well as conformity analysis. Results: The Arabic version of the person-centered climate questionnaire—patient version—showed good reliability as the Cronbach’s alpha value of the total of 17 items was 0.84, and the Cronbach’s alpha values of the three sub-scales (safety, everydayness, and hospitality) were 0.83, 0.56, and 0.68, respectively. Internal consistency results were high in terms cof orrelation coefficient for all 17 items. The exploratory factor analysis identified the three factors (safety, everydayness, and hospitality) responsible for 47.174% of the total variance. Conclusion: The Arabic version of the PCQ-P showed satisfactory reliability and validity for measuring patients’ perceptions of person-centeredness in Arab healthcare settings. This Arabic version will be accessible to those interested in generating and using empirical evidence to promote a patient-centered care approach in Arab healthcare settings. The results of this study can be used as a starting point for assessing and developing a person-centered care culture in Saudi hospitals and other Arab countries in the Middle East.

## 1. Introduction

More than two decades ago, the Institute of Medicine (IOM) included patient-centered care as one of the six essential aims of building the future health care system [1]. The term ‘patient-centered care’ is often used interchangeably with ‘person-centered care’. However, Person-Centered Care (PCC) is commonly used in the literature [2]. The World Health Organization (WHO) has defined person and people-centered care as “one in which individuals, families and communities are served by and are able to participate in trusted health systems that respond to their needs in humane and holistic ways” [3]. The PCC approach aims to achieve better health outcomes by ensuring that healthcare is rooted in universal principles of human rights and dignity, non-discrimination, participation and empowerment, access and equity, and a partnership of equals [4]. Available evidence on PCC showed a positive impact on improving communication, involvement, and relationship with health care providers. Studies also showed that forming value-based relationships between patients and healthcare providers could lead to better adherence to treatment plans [5,6,7,8,9]. 

Nevertheless, Ponte et al. touched on, among other possible factors, the challenges of changing the well-established traditional patterns that have contributed to a PCC setback [10]. Similarly, perceptions of increased cost and time are also perceived as barriers to implementing PCC [11]. Moreover, drawing attention to PCC conceptual issues, Mead and Bower claimed that the vagueness of what PPC means and how it is applied is another obstacle [12]. Thus, one might argue that the scientific utility of patient-centeredness could be seriously compromised if it is not well-defined or measured in a uniform manner. 

On the other hand, published studies on PCC showed that it helps healthcare organizations increase patient satisfaction, improve healthcare quality, and promote patient adherence to treatment plans, but the barriers to implementing PCC must be overcome to achieve those benefits [13]. Thus, several health systems in many high-income countries are moving towards people-centered health care approaches. Yet, PCC is still restricted in low- and middle-income countries, as social, religious, cultural, and economic barriers to their health systems abound, limiting access to quality healthcare. 

There is an increasing number of studies exploring the urgency, feasibility, and strategies of implementing a person-centered approach in low- and middle-income countries’ health systems [14,15,16,17]. Lack of integration and coordination at the healthcare system levels between primary, secondary, and tertiary care in developing countries in general and the Middle East in particular are among the main challenges of the PCC approach. Further, coordination between public and private health care systems should be addressed to facilitate and support the successful delivery of person-centered care [18,19]. 

Based on earlier studies, the implementation of a person-centered care approach in the healthcare sector may demand key transformations in developed and developing nations to improve the quality of healthcare services. However, it has gained little attention among the scientific community and regulatory bodies, which became apparent and started controversies among healthcare service providers, professionals, and decision-policy makers nearly all around the world [20]. Likewise, developed countries in their context established various approaches and tools for the implementation of PCC. In the same vein, the Middle Eastern countries, including Saudi Arabia, are trying to make a move towards the implementation of PCC approaches in their healthcare sector as well. Thus, this study intends to find out the validity and reliability of the translated Arabic version of the PCQ-P among Arabic-speaking patients attending Saudi hospitals, measuring to what extent the climate of health care settings is perceived as being person-centered.

## 2. Methods

This is a psychometric evaluation analysis of the Arabic version of the PCQ-P based on an across-sectional survey conducted to measure the extent to which patients perceive the work environment of healthcare settings as person-centered, and published elsewhere [20]. 

### 2.1. Sampling and Data Collection

Participants were recruited from King Faisal Specialist Hospital and Research Centre, a tertiary hospital in Riyadh, Saudi Arabia. The sample size was calculated using G* power 3.0 software [21]. The G* power package is a commonly used program that depends on Cohen’s power for sample size calculation. Using a power level of 0.90, an alpha level of 0.05, and the smaller effect size (standardized mean difference between small and medium) of 0.20 for a two-tailed independent t-test, the minimum estimated sample size was 255. Hence, the sample size was increased to allow for non-response probabilities and ensure that a large sample is obtained, which will help in providing more valid findings using advanced statistical analysis. The actual sample size comprised 300 patients. 

The inclusion criteria for this psychometric assessment study were based on adult patients admitted to (for more than two days—48 h) anyone of the 16 inpatient departments, including General Surgery; Medical and Surgical Cardiology; Urology; Ear, Nose, and Throat; Gynecology; Hematology; and Internal Medicine. All participants are fluent in Arabic and willing to participate in this study. The exclusion criteria were clinically unstable patients or patients on the pediatric ward. Data were collected by two trained nurses who were not working at any of the 16 different wards over a seven-month period of time. Before the collection of data, these nurses explained the objectives of the study and ensured the confidentiality of the provided information, which was used only for the purpose of this research. The data were collected through face-to-face interviews by the qualified nurses on the day of a patient’s discharge or the day before for those who met the inclusion criteria. 

### 2.2. Study Instruments

The original English version of the Person-Centered Climate Questionnaire-Patient version (PCQ-P), which supports the individual’s personhood in healthcare settings, was developed by a group of researchers in Sweden to measure care environments and compare how patients perceived person-centered care in different inpatient units, as reported by Edvardsson et al. The patient version (PCQ-P) was the initial version of the PCQ instrument, which has since been further developed for other contexts, including the staff version (PCQ-S) and the family member version (PCQ-F) [22,23,24]. 

All versions of PCQ tools were developed and designed from the research literature on PCC and based on qualitative studies exploring healthcare environments that maintain the individual’s personhood by offering a person-centered climate [22,23,25,26]. The PCQ-P consisted of 17 items covering three dimensions (a climate of safety, everydayness, and hospitality). The climate of safety was measured through items 1–10, everydayness through items 11–14, and hospitality through items 15–17. A study conducted by university researchers with a total of 544 hospital patients in Sweden assessed the validity and reliability of the survey’s patient version [22]. Earlier versions of the PCQ-P showed satisfactory reliability, as evidenced by Cronbach’s alpha values of 0.93 and for the subscales (safety, everydayness, and hospitality) with Cronbach’s alpha values of 0.94, 0.82, and 0.64, respectively [22]. The questionnaire items were ranked on a 6-step Likert scale (ranging between 1 = no, I disagree completely to 6 = yes, I agree completely). The total score ranged from 17 (not very person-centered) to 102 (very person-centered). The PCQ-P tool has been applied and tested in many countries, including Australia, the USA, and South Korea [27,28,29]. 

### 2.3. Translation Process and Pilot Study

The Swedish version of the PCQ-P was translated into English by the original author [27]. The tool was translated from English into Arabic using the forward- and back-translation technique. First, bilingual research experts translated the instrument from English into Arabic (forward translation). Second, a specialist in the Arabic language was consulted to modify the translated version and provide an adequate translation. Third, the modified Arabic version was back-translated into English by another independent bilingual who was blind to the original instruments (back-translation). Fourth, both versions were discussed by a panel of five healthcare experts to assess whether the Arabic and English versions were similar to the original questionnaire and identify differences that could have arisen in the different stages of the translation. Lastly, the expert panel addressed all minor discrepancies through consensus discussion and deemed the Arabic version appropriate. The pilot study of 20 participants was conducted to get their feedback on the clarity or ambiguity of each question and estimate the average time needed to complete the survey. Following the pilot study, the internal consistency reliability estimates for the PCQ-Patient version were satisfactory, as the Cronbach’s alpha reached 0.85. Unlike the original questionnaire, the scale was altered from a 6-step Likert scale to a 5-step scale in line with the results of the pilot study and following the recommendations made by the expert panel. 

### 2.4. Statistical Analysis

The Statistical Package for the Social Sciences (SPSS version 22; IBM, NY, USA) was used for statistical analyses. Descriptive statistics with frequencies, percentiles, ranges, means, and standard deviations were also used for each item on the survey to describe the sample and the respondents’ socio-demographic characteristics. The item performance and internal consistency reliability on total scale and subscale levels of the PCQ-P Arabic version were assessed to determine the psychometric characteristics of the tool. Cronbach’s alpha coefficient, item-total correlations, and Cronbach’s alpha if an item is deleted were explored to determine internal consistency. The exploratory factor analysis (EFA) and confirmatory methods were used to assess the construct validity. The Kaiser-Meyer-Olkin (KMO) and Bartlett’s test of sphericity were used to measure sampling adequacy for factor analysis. 

### 2.5. Patient and Public Involvement 

The ultimate purpose and benefit of this study are to provide a valid and reliable tool to measure patient-centeredness in the Arab-speaking context of the health system. Patients were involved as part of this study during the pilot study testing the applicability and feasibility of the translated study tool as well as during the collection of the primary data.

### 2.6. Ethical Aspects

Ethical approval was obtained from the Research Ethics Committee at King Faisal Specialist Hospital and Research Center (reference: NP and R/12/37). Patients meeting the inclusion criteria were asked to participate in this study, and those who verbally agreed to take part signed a written informed consent form secured in their files. The authors maintained and managed complete confidentiality throughout and after the study.

## 3. Results

### 3.1. Sample Descriptions

A total of 300 patients were involved in the psychometric evaluation. The sample consisted of 53% females, and 39.7% of the respondents were aged 21–40 years. Most participants (96.7%) were Saudi, with no health insurance (85.3%), married (72.7%), and unemployed (67.4%). Most participants had either a high school diploma or below (49%) or a bachelor’s degree (32%). Less than one-third of the respondents had no income (29%); about half of the respondents had a monthly income less than the SR; and 24.7% earned more. The length of stay for most patients in the hospital was less than a week, and most (84.3%) preferred to receive treatment in public hospitals (Table 1). 

### 3.2. PCQ-P Item Performance

The total mean score for the PCQ-P Arabic version was 73 ± 9.988 out of 85, indicating that patients perceived their healthcare environments as having a highly person-centered climate of care. The mean score for each item ranged from 2.55 to 4.92. The lowest scoring statement was item number 11, “A place that has something nice to look at (e.g., views and artwork)”, with a mean of 2.55 ± 1.64, as respondents felt that hospital environments generally lacked any form of art, decoration, or scenery. On the other end, the highest-scoring statement was item number 3, “A place where I feel safe” (4.92 ± 0.40), which indicates a safe and orderly environment. The highest mean subscale scores belonged to “a climate of safety”, followed by “a climate of hospitality”, and the lowest mean scores were found in “a climate of everydayness”.

### 3.3. Reliability and Validity

As shown in Table 2, the Cronbach’s alpha value of the PCQ-P as a whole scale was 0.84, indicating that the Arabic version of the PCQ-P had adequate internal consistency and could reliably measure the person-centered climate as perceived by the patient in a tertiary care hospital. The Cronbach’s alpha value for the Arabic PCQ-P was higher than the acceptable alpha value of 0.7 [30].

The overall Cronbach’s alpha values of the three subscales were safety, 0.83; everydayness, 0.56; and hospitality, 0.68; showing that PCQ-P had satisfactory psychometric properties on subscale levels (Table 2). The Alpha Cronbach’s coefficient is greater than 55% for all factors; in addition, the Alpha Cronbach’s coefficient for the total is equal to (0.84) and therefore can be relied on to measure the study variables. Since the Cronbach’s alphas for items that were deleted ranged between 0.81 and 0.84, omitting any item from the questionnaire would not affect the tool’s reliability. The itemized total correlation coefficients had satisfactory values ranging from 0.19 to 0.63. Item number 3, “A place where I feel safe”, had the smallest correlation coefficient of 0.19, which suggests that this item did not correlate well with the total scale, yet excluding it would not affect the overall reliability. 

### 3.4. Explanatory and Confirmatory Factor Analysis

Table 3 and Table 4 show the explanatory factor analysis. Kaiser-Meyer-Olkin (KMO) was performed to measure sampling adequacy. The KMO is equal (0.885), which indicates that the measurement is excellent as the minimum value of that test was (0.6) and the Barlett’s test for sphericity was significant (χ2 = 1453.107, *p* = 0.01). Thus, the data was considered appropriate for factor analysis. The analysis extracted a three-factor solution, each with eigenvalues above (1.1), which explain 47.174% of the total variance. Figure 1 shows the component number with Eigenvalues above (1.1). Table 4 shows that out of the 17 items in the questionnaire, 8 items which accounted for 32.45% of variances and were categorized as the first factor of “safety”, 5 items accounting for 7.79% of variances were loaded on the second factor of “everydayness”, and the remaining four items accounting for 6.92% of variances were categorized under the third factor of “hospitality”. Overall, the final model of principal components analysis (PCA) produced three main loading factors (safety, everydayness, and hospitality) responsible for explaining 47.174% of the total variance. 

In addition, we performed the confirmatory factor analysis of three factors to test the hypothesis about the existence or absence of a relationship between the three factors (safety, everydayness, and hospitality) and a relationship between the items of these factors, and we evaluated the ability of the model to express the data set. Figure 2 shows the confirmatory factor analysis of the three factors. Table 5 shows the model fit summary for the confirmatory factor analysis. The CMIN/DF is equal to (1.775), and the different researchers have recommended using ratios as low as three or as high as five to indicate a reasonable fit [31]. The Goodness of Fit Index (GFI) is equal to 0.924, and this value close to one; a value of one indicates a perfect fit. The Comparative Fit Index (CFI) is equal to 0.936, which means it falls in the range from 0 to 1, and CFI values close to one indicate a very good fit. The root mean square residual (RMR) is equal to 0.053, and RMR values close to zero indicate a good fit, while the value of RMR equal to zero indicates a perfect fit. The Tucker-Lewis coefficient (TLI) is equal to 0.936, and the typical range of this coefficient is between zero and one. TLI values close to one indicate a very good fit. The results show that there is a correlation between the three factors and a relationship between the items of these factors, and all the quality indicators of the model achieved an acceptable level, and some came close to the required level.

## 4. Discussion

The aim of the study was to examine the psychometric properties of the Arabic person-centered Climate Questionnaire Patient version (PCQ-P) at a tertiary care hospital in Saudi Arabia. An extensive review of literature was conducted in the context of this empirical investigation, indicating a gap in knowledge and supporting an evaluation of the content validity and reliability of the Arabic version of the PCQ-P. The findings of this study are consistent with previous evaluations of different language versions in various contexts [22,32]. These studies were conducted in a similar fashion and also support the discovery and advocacy of the argument that was developed in the Saudi context of PCQ-P validity and reliability. The PCQ-P Arabic version had satisfactory reliability and validity, as shown by a Cronbach’s alpha across the total scale and subscales. The Cronbach’s alpha coefficient for the PCQ-P instrument (0.84) revealed that the Arabic version of the PCQ-P also had satisfactory internal consistency, reliability, and validity. 

This internal consistency level of reliability in the present study, which is well aligned with the Swedish hospital study, which reported a Cronbach’s alpha coefficient of 0.93. Moreover, it is also consistent with a Norwegian hospital study, which reported a Cronbach’s alpha value of 0.84 [22,32]. In this study, Cronbach’s alpha values for the three subscales (safety, 0.83; everydayness, 0.56; and hospitality, 0.68) were acceptable compared with the English version (0.88, 0.77, and 0.50, respectively) [27] (Yoon et al. 2015), and the Swedish version as well (safety, 0.94; everydayness, 0.82; and hospitality, 0.64) [28]. 

A literature review was conducted to compare the psychometric analysis and responsiveness of different tools and components on the PCC. This extensive review revealed the significance of examining psychometric properties within hospital settings in Arabic contexts, as no such studies were found. The findings of different studies reported that the PCQ-P version was a suitable and reliable instrument in various contexts [33]. The mean score of the present study was 73, which is lower than the previous studies as it used the five-Likert scale as compared to the six-Likert scale that was used by other studies like the US English version (92.5) and Norwegian version (86.5). Nevertheless, the small difference between the three samples’ mean scores could suggest that cultural diversities and differences are not necessarily highly influencing the PCQ-P. This confirms that the questionnaire measures what it was built for, and all variables are clear and unambiguous.

Many health care systems around the globe have replaced the paternalistic model, where healthcare professionals decide on behalf of patients and their family members, with a more person-centered care practice that considers patients as partners involved in the decision-making process concerning their treatment and needs. Studies have reported several benefits of adopting and implementing PCC into health systems, such as increased patient satisfaction with care, better use of resources, decreased costs, and ultimately improved health outcomes [34,35,36,37,38,39]. Having tools to evaluate perceived levels of person-centeredness is key to scientifically evaluating such benefits and how they can develop over time.

Delivering healthcare that foregrounds partnerships between different stakeholders, including patients, family members, and healthcare professionals, can yield mutual benefits in healthcare organizations as patients and families become more involved and accountable for their care and thus would be expected to rate the care climate as being more person-centered for example with the PCQ-P. There is a growing body of studies that provide evidence for a link between a person-centered approach, improvements in healthcare quality and safety, reductions in costs, and an increase in patient and provider satisfaction [7,40,41,42,43]. The Arabic PCQ-P can contribute further to establishing such data in new contexts as well.

### Limitations

There are some limitations to this study that need to be addressed. First, the results were based on a cross-sectional design that must be considered when interpreting the findings. Second, the participants were only recruited from a single hospital, which may restrict the generalizability of the results. Third, the Arabic tool was solely investigated in Saudi Arabia, and further psychometric evaluation of the scale in different samples and settings in the region is recommended to support the instrument’s reliability and validity in other Arab countries’ settings. 

## 5. Conclusions

The overall results showed satisfactory validity and reliability in assessing patients’ perceptions of PCC in the Saudi hospital setting. The instrument can be used as a tool to measure the extent of person-centeredness from the patient’s perspective in hospitals across Arab countries. Future research is also needed to evaluate PCQ-P in different healthcare organizations in Saudi Arabia.

### Strengths and Limitations of This Study

To the best of our knowledge, this methodological study on the psychometric evaluation of the Arabic version of the Person-Centered Climate Questionnaire-Patient version, is the first to be conducted and published in the Middle East context.The study findings showed a satisfactory level of reliability and validity for the Person-Centered Climate Questionnaire in an Arabic-speaking health care context, which will be a practical tool to measure patient and person-centeredness in Arabic-speaking health care systems. We believe this study will add value for promoting the Person-Centered care (PCC) approach in healthcare in the Middle East and Arabic countries. Among the limitations of the study is that it was conducted in only one country, and its replication in other Arab countries is highly recommended.

## Figures and Tables

**Figure 1 healthcare-11-00190-f001:**
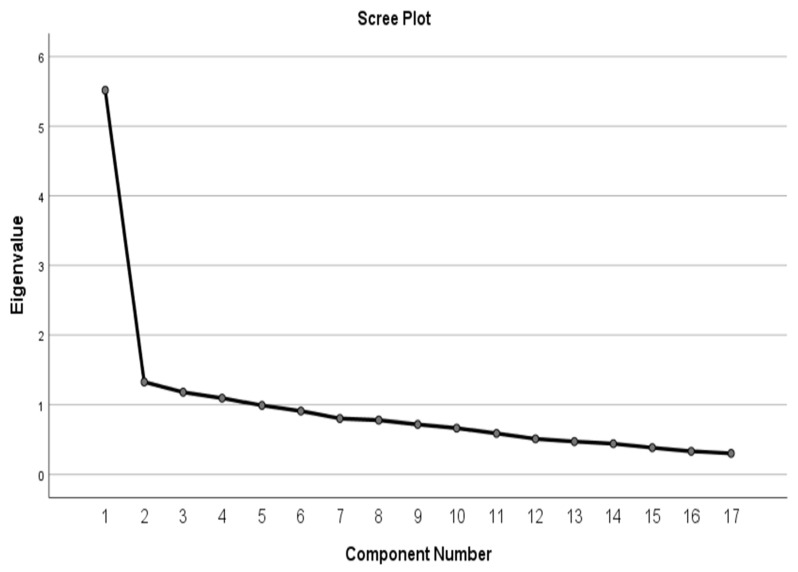
Scree plot chart and extracted factors based on Eigenvalues.

**Figure 2 healthcare-11-00190-f002:**
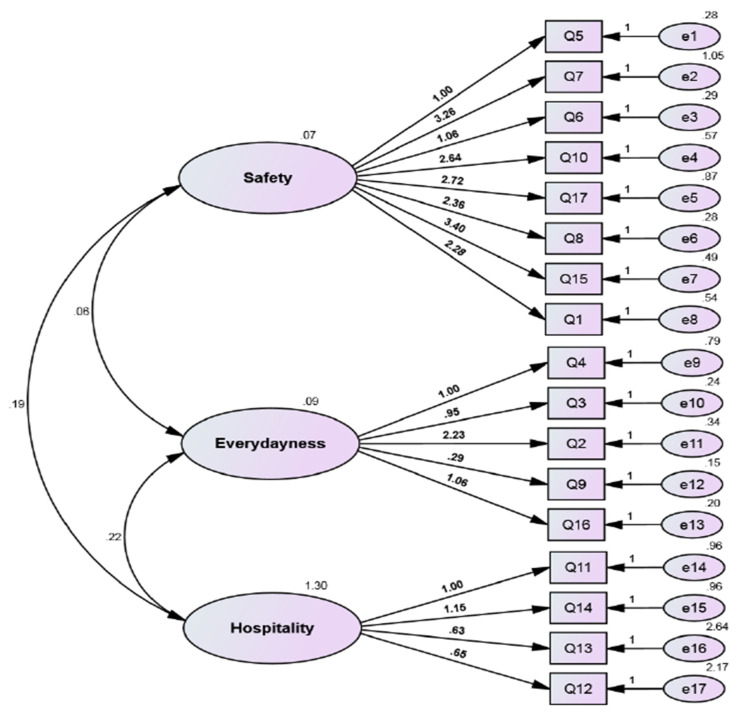
Confirmatory factor analysis of the three factors.

**Table 1 healthcare-11-00190-t001:** Demographic characteristics.

Variables	Frequency	Percentage
**Age (years)**		
≤20	20.0	6.7
21–40	119.0	
41–60	102.0	
≥60	59.0	
**Gender**		
Male	141	47
Female	159	53
**Nationality**		
Saudi	290.0	96.7
Non-Saudi	10.0	3.3
**Health insurance**		
Yes	44.0	14.7
No	255.0	85.3
**Marital status**		
Married	218.0	72.7
Unmarried	82.0	27.3
**Educational level**		
Illiterate	47.0	15.7
High school and below	147.0	49.0
Bachelor	96.0	32.0
Postgraduate	10.0	3.3
**Occupation**		
Employed	97.0	32.6
Unemployed	201.0	67.4
**Monthly income (SR)**		
No income	88.0	29.7
<5000	61.0	20.6
5000–10,000	74.0	25.0
>10,000	73.0	24.7
**Area of residency**		
Riyadh	118.0	39.3
Outside Riyadh	182.0	60.7
**Area of admission**		
Emergency department	91.0	30.4
Elective case or outpatient appointment	119.0	39.8
Referral from another hospital	89.0	29.8
**Duration of hospitalization (weeks)**		
<1	135.0	45.0
1–2	79.0	26.3
>2	86.0	28.7
**Hospital preference for treatment**		
Governmental	252.0	84.3
Teaching	6.0	2.0
Private	41.0	13.7

**Table 2 healthcare-11-00190-t002:** Item performance and reliability estimates of the PCQ-P (*n* = 300).

PCQ-P/Item ^	Mean ± SD	Alpha If Item Deleted	Correlation with Total
**Safety** (*Cronbach’s Alpha:* **0.83**)			
1. A place where staff are knowledgeable	4.76 ± 0.59	0.83	0.38
5. A place where it is easy to talk to staff	4.56 ± 0.94	0.82	0.51
6. A place where staff take notice of what I say	4.67 ± 0.80	0.82	0.63
7. A place where staff come quickly when I need them	4.34 ± 1.13	0.81	0.67
8. A place where staff talk to me so that I can understand	4.78 ± 0.60	0.83	0.39
10. A place where staff seem to have time for patients	4.33 ± 1.16	0.83	0.48
15. A place where staff make extra efforts for my comfort	4.18 ± 1.33	0.82	0.57
17. A place where I can get that ‘little bit extra’	4.49 ± 1.01	0.82	0.58
**Everydayness** (*Cronbach’s Alpha:* **0.56**)			
2. A place where I receive the best possible care	4.61 ± 0.89	0.82	0.58
3. A place where I feel safe	4.92 ± 0.40	0.84	0.19
4. A place where I feel welcome	4.84 ± 0.52	0.83	0.43
9. A place that is neat and clean	4.86 ± 0.55	0.83	0.36
16. A place where I can make choices (e.g., what to wear, eat, etc.)	4.60 ± 0.93	0.84	0.24
**Hospitality** (*Cronbach’s Alpha:* **0.68**)			
11. A place that has something nice to look at (e.g., views, artwork, etc.)	2.55 ± 1.64	0.84	0.38
12. A place that feels homely	3.89 ± 1.50	0.82	0.56
13. A place where it is possible to get unpleasant thoughts out of your head	3.67 ± 1.63	0.82	0.59
14. A place where people talk about everyday life and not just illness	2.96 ± 1.77	0.84	0.35

^ Total Cronbach’s alpha: **0.84.**

**Table 3 healthcare-11-00190-t003:** Initial Eigenvalues, total variance explained, and cumulative percentage of the three primary factors extracted from the PCQ-P.

Component	Initial Eigenvalues			Rotation Sums of Squared Loadings		
	Total	% of Variance	Cumulative %	Total	% of Variance	Cumulative %
1	5.516	32.450	32.450	3.730	21.943	21.943
2	1.325	7.796	40.246	2.180	12.821	34.764
3	1.178	6.928	47.174	2.110	12.410	47.174
4	1.092	6.426	53.599			
5	0.988	5.814	59.414			
6	0.907	5.335	64.749			
7	0.801	4.713	69.461			
8	0.778	4.576	74.037			
9	0.717	4.219	78.257			
10	0.665	3.913	82.170			
11	0.589	3.464	85.634			
12	0.511	3.007	88.640			
13	0.472	2.777	91.418			
14	0.441	2.595	94.013			
15	0.384	2.257	96.270			
16	0.332	1.954	98.223			
17	0.302	1.777	100.000			

**Table 4 healthcare-11-00190-t004:** Rotated Principal Component Matrix of PCQ-P.

Items		Components		
		Factor 1 Safety	Factor 2 Everydayness	Factor 3 Hospitality
5	A place where it is easy to talk to staff.	0.721	−0.034	0.184
7	A place where staff come quickly when I need them.	0.717	0.189	0.299
6	A place where staff takes notice of what I say.	0.669	0.355	0.161
10	A place where staff seem to have time for patients.	0.653	0.102	0.131
17	A place where I can get that ‘little bit extra’.	0.626	0.205	0.244
8	A place where staff talk to me so that I can understand.	0.612	0.087	−0.024
15	A place where staff make extra efforts for my comfort.	0.600	0.157	0.296
1	A place where staff are knowledgeable.	0.426	0.398	−0.053
4	A place where I feel welcome.	0.363	0.637	−0.072
3	A place where I feel safe	−0.042	0.557	0.023
2	A place where I receive the best possible care.	0.480	0.515	0.194
9	A place that is neat and clean.	0.169	0.495	0.180
16	A place where I can make choices (e.g., what to wear, eat, etc).	0.072	0.432	0.113
11	A place that has something nice to look at (e.g., views, artwork, etc).	0.151	−0.018	0.707
14	A place where people talk about everyday life and not just illness.	0.143	−0.001	0.652
13	A place where it is possible to get unpleasant thoughts out of your head.	0.217	0.394	0.637
12	A place that feels homely.	0.160	0.452	0.613

**Table 5 healthcare-11-00190-t005:** The quality indicators of the model.

Model Fit Summary	
Measure	Estimate
CMIN	205.953
DF	116
CMIN/DF	1.775
GFI	0.924
CFI	0.936
TLI	0.925
RMR	0.053

## Data Availability

The data presented in this study are available on request from the corresponding author.

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
