# Peer review of "Psychometric Evaluation of the Arabic Version of the Person-Centered Climate Questionnaire: Patient Version (PCQ-P)"

_healthcare, 2023, doi:10.3390/healthcare11020190_

Round 1
Reviewer 1 Report
Overview and general recommendation
After reading and reviewing the manuscript, I note that it is a continuation of a research line that seeks to understand the state of person-centred care in different international contexts. It is interesting that the questionnaire, on this occasion, is adapted to a different cultural setting from previous ones. It is found to be quite well adapted in the Middle East despite the limitations of the study. The research is well conducted in all its sections, with the statistical analysis standing out, which is the result of a methodology and use of the questionnaire that has been consolidated over the last few years.
Major comments
Although there are no major comments as such, there are three aspects that I would like to see answered:
- Were cut-off points adapted to the 5-step Likert scale version of the questionnaire established ("not very person-centred climate" and "very person-centred climate")? If yes, they should be added in the manuscript.
- Regarding the rotated matrix, have you tried other types of rotations to check whether items 1 and 2 further differentiate their results between factor 1 and 2?
- In the Author Contributions, DE's participation is not specified, when he is listed as one of the co-authors of the research. What was his participation?
Minor comments
- Throughout the manuscript: (1) It is necessary to unify whether you use the English term (centre) or the American term (centre) for this word. (2) There are many double spaces in many lines. (3) Reference numbers should be in square brackets and not in superscript.
- Page 1, line 10: Please reduce the font size of the affiliation.
- Page 1, line 11: The comma and full stop at the end should be removed.
- Page 1, lines 41-43: A reference needs to be added to this citation.
- Page 2, lines 51 and 55: The reference style stated by the journal is not APA. Please remove the year of publication from these citations. Please review this in line 261 on page 10.
- Page 3, line 104: I think there is a typo in the middle of line (p).
- Page 3, line 111: The line break should be removed.
- Table 1: Please revise the total percentage of the variable "Year", as the sum results in 100.1%.
- Table 1: Please replace "Gender" with "Sex", as you are referring to the biological variable and not psychosocial or cultural. You can find more information on this at the following web link: https://www.mdpi.com/journal/healthcare/instructions#ethics
- Table 2: Please add a space between the sign and the standard deviation in items 7 and 9.
- Page 6, line 214: Please replace "figure" with "Figure". I add the same comment for "table" at the end of the line.
- Page 6, line 220: Please check the italics in the text in brackets.
- Figure 2: In the title, replace the colon with a full stop. Also, there is a typo in "Confirmatroy" and "the" is duplicated.
- Table 5: Replace "the" with "The".
- Page 11, line 328: The "No" is sufficient information.
- References: References should be adapted to the Healthcare's instructions. For example: (1) The year of publication of journal articles should be in bold, (2) the volume of journals should be in italics, (3) there should be a comma and a space before the page ranges of articles published in journals, not a colon. Please also check references number 13, 14, 19, 37, and 41 more completely.
I hope my comments will help you to improve the manuscript.
Author Response
Comments and Suggestions for Authors
Overview and general recommendation
After reading and reviewing the manuscript, I note that it is a continuation of a research line that seeks to understand the state of person-centred care in different international contexts. It is interesting that the questionnaire, on this occasion, is adapted to a different cultural setting from previous ones. It is found to be quite well adapted in the Middle East despite the limitations of the study. The research is well conducted in all its sections, with the statistical analysis standing out, which is the result of a methodology and use of the questionnaire that has been consolidated over the last few years.
Reply: Thanks for your nice comments
Major comments
Although there are no major comments as such, there are three aspects that I would like to see answered:
- Were cut-off points adapted to the 5-step Likert scale version of the questionnaire established ("not very person-centred climate" and "very person-centred climate")? If yes, they should be added in the manuscript.
Reply: The cut-off points did not adapt in the modified questionnaire on 5-step Likert scale to suit the study context based on the feedback got from the pilot study testing the reliability and content validity of the study toll
- Regarding the rotated matrix, have you tried other types of rotations to check whether items 1 and 2 further differentiate their results between factor 1 and 2?
Reply: Yes, we did, attempting other types of rotations and observed very much similar results between the factors. Moreover, the authors performed the confirmatory factor analysis to find out any kind of inconsistencies or deviations in any items of the three factors.
- In the Author Contributions, DE's participation is not specified, when he is listed as one of the co-authors of the research. What was his participation
Reply: The contributions of the third author (DE) and his participation have been added to the revised version
Minor comments
- Throughout the manuscript: (1) It is necessary to unify whether you use the English term (centre) or the American term (centre) for this word. (2) There are many double spaces in many lines. (3) Reference numbers should be in square brackets and not in superscript.
Reply: Thanks for your comments, it will be amended in the revised version
- Page 1, line 10: Please reduce the font size of the affiliation.
Reply: Done as shown the revised version
- Page 1, line 11: The comma and full stop at the end should be removed.
Reply: removed as shown in the revised version
- Page 1, lines 41-43: A reference needs to be added to this citation.
Reply: This concern has been addressed in the revised version of the manuscript
- Page 2, lines 51 and 55: The reference style stated by the journal is not APA. Please remove the year of publication from these citations. Please review this in line 261 on page 10.
Reply: Thanks for your comment, we deleted the year of the publication as shown in the revised version
- Page 3, line 104: I think there is a typo in the middle of line (p).
Reply: Thanks, fixed and rephrased as shown in the revised version
- Page 3, line 111: The line break should be removed.
Reply: The line break was removed as shown in the revised version
- Table 1: Please revise the total percentage of the variable "Year", as the sum results in 100.1%.
Reply: Done as shown the revised version
- Table 1: Please replace "Gender" with "Sex", as you are referring to the biological variable and not psychosocial or cultural. You can find more information on this at the following web link: https://www.mdpi.com/journal/healthcare/instructions#ethics
Reply Thanks for your important comments. Actually, we prefer keeping “Gender” rather than “Sex” to also reflect the cultural difference rather than biological difference in this study
- Table 2: Please add a space between the sign and the standard deviation in items 7 and 9.
Reply: Done throughout Table 2 as shown in the revised version
- Page 6, line 214: Please replace "figure" with "Figure". I add the same comment for "table" at the end of the line.
Reply: Done as shown the revised version
- Page 6, line 220: Please check the italics in the text in brackets.
Reply: Done as shown the revised version
- Figure 2: In the title, replace the colon with a full stop. Also, there is a typo in "Confirmatroy" and "the" is duplicated.
Reply: Done as shown the revised version
- Table 5: Replace "the" with "The".
Reply Done as shown in the revised version
- Page 11, line 328: The "No" is sufficient information.
Reply: Fixed as shown in the revised version…
Reviewer 2 Report
Congratulations. It is a work that deals with a novel topic of scientific and health interest.
However, the article requires a number of revisions prior to publication.
-It is not necessary for the abstract to be divided into sections. The reading flow will be more attractive without divisions.
- Keywords should be reviewed and check that MESH or DeCS terms were used.
-In the introduction, the rationale for the paper should be presented more concisely.
-In the methodology, I would expand the information on data collection and the interviews conducted by the two nurses, explain in more detail the statistical analysis section and add a section to discuss the ethical aspects of the study.
-The results section provides interesting information but the organisation is very chaotic. The results should be presented in a more orderly fashion. Review the structure of this section, the arrangement of the tables and figures, the explanatory paragraphs and the relationship between them.
- The discussion aims to compare the results of this study with the rest of the literature published on the subject. On occasions, some paragraphs belong to the introduction rather than to the discussion. Revise this aspect.
- There are problems with the bibliographic style used in this article as it does not meet the requirements and guidelines of the reference style of this journal.
Author Response
Comments and Suggestions for Authors
Congratulations. It is a work that deals with a novel topic of scientific and health interest.
However, the article requires a number of revisions prior to publication.
-It is not necessary for the abstract to be divided into sections. The reading flow will be more attractive without divisions.
Reply: Thanks for your feedback and sure will follow the journal instructions in this regard
- Keywords should be reviewed and check that MESH or DeCS terms were used.
Reply: Sure, we think the keywords are well-matched with MESH the revised version
-In the introduction, the rationale for the paper should be presented more concisely.
Reply: we appreciate reviewer comments regarding the concise rationale of the study. This concern has been addressed in the last paragraph in the introduction section as shown in the revised version of the manuscript.
-In the methodology, I would expand the information on data collection and the interviews conducted by the two nurses, explain in more detail the statistical analysis section and add a section to discuss the ethical aspects of the study.
Reply…The highlighted issues by the reviewer have been addressed and precise explanations incorporate in the revised version of the manuscript.
-The results section provides interesting information, but the organisation is very chaotic. The results should be presented in a more orderly fashion. Review the structure of this section, the arrangement of the tables and figures, the explanatory paragraphs and the relationship between them.
Reply: Thank you to the reviewer – to draw our attention to the organizations, structures, and arrangements of various tables and figures. This concern has been addressed by authors in revised versions of the manuscript.
- The discussion aims to compare the results of this study with the rest of the literature published on the subject. On occasions, some paragraphs belong to the introduction rather than to the discussion. Revise this aspect.
Reply: Thank you for this comment. We have now scrutinized the discussion section and made some edits to improve clarity and argumentative rigor. We believe the discussion achieves what it aims to achieve, discussing the findings of our study in relation to existing studies that exist and were highlighted in the introduction. Hence, some sections will inevitably tie together introductory paragraphs with our findings, and therefore we believe these sections indeed are better placed in the discussion section.
Round 2
Reviewer 2 Report
Most of the requested revisions have been implemented. The article is suitable for publication in its current form.